# Integrated care in patients with atrial fibrillation- a predictive heterogeneous treatment effect analysis of the ALL-IN trial

Emmy M. Trinks-Roerdink[1]*, Geert-Jan Geersing[1], Carline J. van den Dries[1], Martin E. W. Hemels[2,3], Michiel Rienstra[4], Isabelle C. van Gelder[4], Maarten van Smeden[5], David van Klaveren[6,7], David M. Kent[7], Frans H. Rutten[1], Sander van Doorn[1]

1 Department of General Practice & Nursing Science, Julius Center for Health Sciences and Primary Care, University Medical Center Utrecht, Utrecht University, Utrecht, the Netherlands, 2 Department of Cardiology, Rijnstate, Arnhem, the Netherlands, 3 Department of Cardiology, Radboud University Medical Center, Nijmegen, the Netherlands, 4 Department of Cardiology, University Medical Center Groningen, University of Groningen, Groningen, Netherlands, 5 Department of Epidemiology & Health Economics, Julius Center for Health Sciences and Primary Care, University Medical Center Utrecht, Utrecht University, Utrecht, the Netherlands, 6 Department of Public Health, Erasmus MC University Medical Center, Rotterdam, the Netherlands, 7 Predictive Analytics and Comparative Effectiveness Center, Tufts Medical Center, Boston, MA, United States of America

* e.m.roerdink@umcutrecht.nl

**Data Availability Statement:** All relevant summary data are provided in the paper and its Supporting

## Abstract

### Introduction

Integrated care is effective in reducing all-cause mortality in patients with atrial fibrillation (AF) in primary care, though time and resource intensive. The aim of the current study was to assess whether integrated care should be directed at all AF patients equally.

### Methods

The ALL-IN trial (n = 1,240 patients, median age 77 years) was a cluster-randomized trial in which primary care practices were randomized to provide integrated care or usual care to AF patients aged 65 years and older. Integrated care comprised of (i) anticoagulation monitoring, (ii) quarterly checkups and (iii) easy-access consultation with cardiologists. For the current analysis, cox proportional hazard analysis with all clinical variables from the $CHA_2DS_2$-VASc score was used to predict all-cause mortality in the ALL-IN trial. Subsequently, the hazard ratio and absolute risk reduction were plotted as a function of this predicted mortality risk to explore treatment heterogeneity.

### Results

Under usual care, after a median of 2 years follow-up the absolute risk of all-cause mortality in the highest-risk quarter was 31.0%, compared to 4.6% in the lowest-risk quarter. On the relative scale, there was no evidence of treatment heterogeneity (p for interaction = 0.90). However, there was substantial treatment heterogeneity on the absolute scale: risk reduction in the lowest risk- quarter of risk 3.3% (95% CI -0.4 - 7.0) compared to 12.0% (95% CI 2.7 - 22.0) in the highest risk quarter.

Information files. Part of the original data used in this study was collected during routine care and is formally owned by The Julius General Practitioners Network, JGPN [https://portal.juliuscentrum.nl/research/nl-nl/cohortsandprojects/cohortsprojects/juliusgeneralpractitionersnetwork/jhn.aspx]. As the owner of the data, JGPN did not provide permission to make the data publicly available due to ethical and legal restrictions. The de-identified data are traceable to the participating centers who did not explicitly approve of raw data publication. However, all interested readers may request data without restriction from JGPN (SecretariaatJHN-3@umcutrecht.nl).

**Funding:** The author(s) received no specific funding for this work.

**Competing interests:** I have read the journal's policy and the authors of this manuscript have the following competing interests: G.J. Geersing, F.H. Rutten, and M.E.W. Hemels report unrestricted institutional grants for performing research in the field of atrial fibrillation from Boehringer-Ingelheim, Bayer Healthcare, BMS Pfizer and Daiichi Sankyo. I. C. van Gelder reports consultancy fees from Boston, BMS and Bayer to the institution, unrestricted research grants from the Netherlands Cardiovascular Research Initiative, unrestricted research grant from the European Union's Horizon 2020 research and innovation programme under grant agreement: EHRA-PATHS (945260). M. Rienstra reports Consultancy fees from Bayer, Microport, InCarda Therapeutics to the institution, an unrestricted research grant from ZonMW and the Dutch Heart Foundation; DECISION project 848090001, unrestricted research grants from the Netherlands Cardiovascular Research Initiative: an initiative with support of the Dutch Heart Foundation; RACE V (CVON 2014–9), RED-CVD (CVON2017-11), an unrestricted research grant from Top Sector Life Sciences & Health to the Dutch Heart Foundation (PPP Allowance; CVON-AI (2018B017)), and an unrestricted research grant from the European Union's Horizon 2020 research and innovation programme under grant agreement; EHRA-PATHS (945260). S. van Doorn reports an unrestricted institutional grant for performing research in the field of stroke diagnosis from Stoffels-Hornstra. This does not alter our adherence to PLOS ONE policies on sharing data and materials.

## Conclusion

While the relative degree of benefit from integrated AF care is similar in all patients, patients with a high all-cause mortality risk have a greater benefit on an absolute scale and should therefore be prioritized when implementing integrated care.

## Introduction

The increasing prevalence of atrial fibrillation (AF) and associated morbidity and mortality have heightened the need for optimizing care of AF patients [1]. The latest guidelines on AF management by the European Society of Cardiology (ESC) recommend integrated AF care by means of the ABC approach, which should entail A) Anticoagulation/Avoid stroke, B) Better symptom control, and C) detection and management of Comorbidities in a multidisciplinary setting (Class IIa recommendation, level of evidence B) [2]. Studies on the clinical effects of integrated AF care have been mainly performed in patients seen in AF clinics, reporting mixed findings. Some studies showed a reduction in adverse events (e.g., a reduction in (cardiovascular) mortality and (cardiovascular) hospital admissions) [3–6], while other studies did not [7–9]. More recently, integrated AF care was studied in Dutch primary care in ALL-IN cluster-randomized trial which demonstrated a large average relative reduction in all-cause mortality by 45% of those in the intervention group compared to usual care [10]. This undisputed benefit notwithstanding, integrated care is a time and resource-intensive intervention. Since the prevalence of AF is expected to increase further in our aging society, and our healthcare system is already under pressure, careful evaluation of which patients should be prioritized when implementing integrated AF care, is of great importance.

To study differences in treatment effects in randomized trials it is common to perform subgroup analyses on predefined subgroups. However, these conventional subgroup analyses have limitations, including the risk of false negative results from lack of power and the risk of false positive results due to multiplicity. Further, because patients differ on so many variables that may influence the outcome of interest and the degree of benefit, results from one-variable-at-time subgroup analysis do not yield patient-centered treatment effect estimates [11]. More recently, a "risk modeling approach" to study heterogeneous treatment effects (HTE) has been recommended to partially address some of the limitations of conventional subgroup analysis. In this approach, a multivariable regression risk model, which takes into account multiple patient characteristics simultaneously, is used to examine how treatment effects vary at different levels of risk for the primary outcome [11, 12]. This study aims to assess whether integrated care should be directed at all AF patients equally by performing a predictive HTE analysis among primary care patients participating in the ALL-IN cluster randomized trial.

## Methods

For this study, we followed the recommendations for HTE analysis stated in the *Predictive Approaches to Treatment Heterogeneity* (PATH) statement [12]. The TRIPOD guideline was used as reporting guideline for predictive studies [13].

### ALL-IN trial

In short, the ALL-IN trial was a cluster-randomized trial in which primary care practices were randomized to provide either integrated AF care or usual care to patients aged 65 years and

older. Highly similar to the ABC approach recommended by the ESC guidelines, our integrated AF care intervention comprised of (i) anticoagulation monitoring in primary care, (ii) quarterly checkups for AF symptoms and comorbidities with special attention for the development of heart failure, and (iii) easy-access consultation with AF- and anticoagulation specialists. In total 26 practices with in total 1,240 patients were included between 2015 and 2017 and the follow-up duration was at least two years. The study design and results of the trial have been described in more detail previously [10, 14].

## Outcome definition

The outcome of this current study is all-cause mortality. This outcome was chosen since the primary outcome of the main study was also all-cause mortality and because HTE analysis is considered only valuable when an overall effect of an intervention is found [12]. Since the ALL-IN trial found an overall effect regarding its primary outcome all-cause mortality, this outcome was selected for the current analysis. Of note, this HTE analysis was not a pre-planned analysis since we initially did not expect to observe superiority of the intervention. In fact, the ALL-IN trial was originally designed as a non-inferiority trial aiming to demonstrate that integrated AF care could be safely orchestrated in a primary care setting [14].

## Model development and internal validation

Although the guideline-recommended $CHA_2DS_2$-VASc (Congestive heart failure, Hypertension, Age, Diabetes, prior Stroke, Vascular disease and Sex) score is widely used to predict stroke in patients with AF, no guideline-recommended prediction model exists for predicting all-cause mortality in AF patients that has shown good performance on an external dataset. Therefore, we developed a new prediction model for the outcome all-cause mortality in a dataset external to the ALL-IN trial. This dataset for model development (derivation cohort) contained data from another cluster-randomized trial performed in primary care in the Netherlands, in which automated $CHA_2DS_2$-VASc decision support for general practitioners regarding treatment with anticoagulants in established patients with AF was studied against usual primary care [15]. The primary outcome of this study was the composite of stroke, TIA, and/or thromboembolism. Data on mortality was also recorded. The inclusion of practices took place between 2013 and 2014 and the follow-up duration of the study was at least two years for every patient. The design and results of this study have been described in more detail previously [15].

Common, well-studied prognostic factors for stroke in patients with AF collected in the $CHA_2DS_2$-VASc score were selected *a priori* as candidate predictors for the model developed. A Cox proportional hazard model was fitted to predict the outcome all-cause mortality over the complete follow-up period of approximately two years, accounting for clustering in primary care practices by adding a random effects term for primary care practice to the model. Both data from the intervention and control groups of the derivation cohort were used for model development. All candidate predictors were added to the model at once, and no predictors were removed from the model. Based on an assumed R-squared between 0.1 and 0.2, an event fraction of 0.11 (261 deaths in 2,355 AF patients), a median follow-up of 2.7 years, and 2 years as the time point of interest for the risk predictions using the Riley minimal sample size criteria, the available sample size was considered sufficient for developing a model with eight parameters. To account for possible non-linearity of age a restricted cubic spline with 3 knots was used. The model was internally validated using bootstrapping with 1000 repetitions to correct for optimism. Discrimination was assessed by calculating Uno's c-statistic with 95% CI,

which is the recommended approach for the validation of survival data [16], and calibration was assessed by creating a calibration plot.

## External validation

The model to predict all-cause mortality was then externally validated in all AF patients participating in the ALL-IN study, both those who received integrated care as well as those who received usual care. The complete follow-up period of two years was used for external validation. To assess the predictive performance of the model in the ALL-IN trial, calibration was determined by creating a calibration plot. Discrimination was assessed by calculating Uno's c-statistic with 95% CI.

## Missing data

In both datasets, predictor variables age and sex, there were no missing data, for the remaining predictors indicating disease history or comorbidity, were considered present in patients in which the electronic file contained evidence of a respective diagnosis, and not present if the electronic file did not report a diagnosis, thus missing data did (strictly speaking) not occur for these data in both datasets.

## Descriptive statistics

Descriptive statistics were used to describe the total study population of the ALL-IN study and the intervention and control group separately, with a mean with standard deviation (SD) or median with an interquartile range (IQR) for continuous variables, and proportions for categorical variables.

**Analysis of heterogeneous treatment effects.** First, a c-for-benefit with 95% CI was calculated. The c-for-benefit is a concordance statistic expressing the probability that from two randomly chosen matched patient pairs with unequal pairwise observed benefit, the pair with greater pairwise observed benefit also has a higher predicted benefit [17]. Any value over 0.5 indicates evidence for treatment heterogeneity. Next, the distribution of the predicted risk of all-cause mortality in the ALL-IN trial was reported by calculating a mean with SD or median with IQR for the total study population, and for the intervention and control group separately. This risk distribution was also graphically assessed. Subsequently, four predefined risk strata were created, and treatment effects were reported across these risk strata. To assess the relative effects of the trial, the hazard ratio for the intervention was plotted as a function of the predicted all-cause mortality risk. To assess the absolute effects the absolute risk reduction was plotted as a function of the predicted all-cause mortality risk. Both absolute and relative treatment effects were plotted as a function of the continuous risk. All plots included a smooth curve, using a spline with 4 degrees of freedom. Finally, to test the null hypothesis (i.e., there is no treatment heterogeneity) on a relative scale, the interaction between treatment and predicted risk was tested for significance. We performed two sensitivity analyses. First, we repeated the HTE analysis using an extended prediction model additionally including chronic kidney disease, COPD and liver disease. Second, we used the $CHA_2DS_2$-VASc score as a continuous numeric univariate predictor of mortality. All statistical analyses were performed in R version 4.2.2.

## Ethics

This is a post-hoc analysis of the ALL-IN trial that received ethical approval of the Medical Ethics Committee of the University Medical Center Utrecht, Utrecht University. In the

**Table 1. Baseline characteristics of the total ALL-IN study population and intervention and control group separately.**

|  | Total (N = 1240) | Intervention group (N = 527) | Control group (N = 713) |
|---|---|---|---|
| Median age (IQR) | 77.0 (11) | 76.0 (10) | 78.0 (11) |
| Median CHA$_2$DS$_2$-VASC score (IQR) | 3.00 (2) | 3.00 (1) | 3.00 (2) |
| Female sex | 613 (49.4) | 239 (45.4) | 374 (52.5) |
| Hypertension | 700 (56.5) | 311 (59.0) | 389 (54.6) |
| Heart failure | 208 (16,8) | 72 (13.7) | 136 (19.1) |
| Diabetes | 316 (25.5) | 131 (24.9) | 185 (25.9) |
| Prior stroke/TIA | 179 (14.4) | 84 (15.9) | 95 (13.3) |
| Coronary artery disease | 213 (17.2) | 93 (17.6) | 120 (16.8) |
| Prior myocardial infarction | 86 (6.9) | 36 (6.8) | 50 (7.0) |
| Peripheral artery disease | 84 (6.8) | 36 (6.8) | 48 (6.7) |
| Prior venous thromboembolism | 55 (4.4) | 25 (4.7) | 30 (4.2) |
| Renal insufficiency | 169 (13.6) | 59 (11.2) | 110 (15.4) |
| COPD | 172 (13.9) | 73 (13.9) | 99 (13.9) |
| History of cancer | 226 (18.2) | 95 (18.0) | 131 (18.4) |

Data are numbers (percentage) unless stated otherwise. IQR: interquartile range. CHA$_2$DS$_2$-VASc: (Congestive heart failure, Hypertension, Age, Diabetes, prior Stroke, Vascular disease and Sex). TIA: transient ischemic attack. COPD: chronic obstructive pulmonary disease.

ALL-IN trial, all eligible patients from practices randomized to the intervention were informed on study purposes and asked for written informed consent before undergoing the intervention. The Medical Ethics Committee provided a waiver of informed consent for the collection of anonymized baseline and outcome data for all eligible patients in both arms, yet all strictly under the auspices of the treating GP. The authors had no access to information that could identify individual participants during or after data collection.

## Results

The study population of the ALL-IN trial consists of 1,240 AF patients (median age 77, IQR 11 years, and 49.4% females); 527 patients in the intervention arm, and 713 patients in the control arm. The baseline characteristics of these patients are presented in Table 1. The mean duration of follow-up was 2.0±0.5 years. In total 135 (10.8%) patients died (incidence rate (IR) of all-cause mortality 5.3 [95% CI 4.4–6.2] per 100 person-years). In the intervention group, 39 patients died (7.4%, IR 3.5 [95% CI 2.5–4.7] per 100 person-years), and in the control group 96 patients (13.5%, IR 6.7 [95% CI 5.4–8.2] per 100 person-years).

### Development, internal- and external validation of the AF prediction model

The study population for model development consisted of 2,355 AF patients (median age 77, IQR 16 years). See Table in S1 Table for the patient characteristics of study population for model development and the ALL-IN study population. See Table in S2 Table for the Cox regression coefficients of the model. The c-index of this model was 0.72 [95% CI 0.69–0.75] at internal validation. The c-index of the externally validated model in the ALL-IN study population was 0.72 [95% CI 0.66–0.78]. See Fig in S1 Fig for the calibration plots in the study population for model development and the ALL-IN population. In both, most patients had a high probability of survival (i.e. a low risk of mortality) which was mostly overestimated by the prediction model. Calibration was best mainly for the higher probabilities of survival (i.e. lower risk of mortality), for which we have the most observations.

## Analysis of heterogeneous treatment effects

The c-for-benefit was 0.58 (95% CI 0.51–0.65). The distribution of the predicted risk of all-cause mortality in the ALL-IN study population is presented in Fig in S2 Fig for the total population, the intervention group, and the control group. At baseline, at the inception of the study cohorts, the median predicted risk of all-cause mortality during approximately two years of follow-up for the total study population was 0.08 (IQR 0.08), for the intervention group 0.07 (IQR 0.07) and for the control group 0.09 (IQR 0.09). Under usual care, the absolute risk of all-cause mortality in the highest-risk quarter was 31.0%, compared to 4.6% in the lowest-risk quarter. Fig 1 shows the event rate, hazard ratio and absolute risk reduction as a function of the predicted all-cause mortality risk, for both the intervention and control group. This figure shows that the event rate is lower for the intervention group compared to the usual care group across all predicted all-cause mortality risk levels. The hazard ratio for the intervention group, compared to the usual care group is constant across all risk levels. The interaction term between the linear predictor and the intervention was not statistically significant (p = 0.90). There was substantial effect heterogeneity on the absolute scale: risk difference in the lowest risk- quarter of risk 3.3% (95% CI -0.4% - 7.0) compared to 12.0% (95% CI 2.7% - 22.0) in the highest risk quarter. Repeating the analyses using the extended model additionally including chronic kidney disease, COPD and liver disease or the $CHA_2DS_2$-VASc score as a univariate predictor did not substantially change the results, see Fig in S3 and S4 Figs.

As an illustration, a 71-year-old, male AF patient with a history of diabetes and vascular disease has a predicted two-year all-cause mortality risk of 7.6%, which corresponds with an HR of 0.49 and an absolute risk reduction of 2.8% during two years of follow-up, comparing integrated AF care to usual primary care: from 7.6% to 4.8%, number needed to treat = 36 for two years. An 89-year-old male patient with a history of diabetes, vascular disease, and heart failure has a predicted all-cause mortality risk of 29.6% in two years, which corresponds with an HR of 0.63 and an absolute risk reduction of 9.3%: from 29.6% to 20.3%, number needed to treat = 11 for two years.

## Discussion

In this additional analysis of the ALL-IN study in elderly AF patients, we evaluated heterogeneity in the effect of integrated AF care in the primary care setting across all-cause mortality risk levels. We showed that *on a relative scale* all patients, independent of baseline all-cause mortality risk, seem to benefit similarly from integrated AF care. On an absolute scale, however, we show that the benefit of integrated care is substantially lower for low-risk patients. Importantly, patients with a higher predicted risk of all-cause mortality (based on easily ascertainable clinical variables from the $CHA_2DS_2$-VASc score) had the greatest absolute risk reduction. Therefore, spending (limited) healthcare resources and time predominantly on integrated care for high-risk patients and applying a more lenient approach to low-risk patients, could be a promising strategy for efficiently managing the increasing healthcare burden associated with the ongoing AF epidemic.

## Comparison with existing literature

While many studies have reported on the effect of integrated care on all-cause mortality, including two systematic reviews and meta-analyses [5, 6], this is the first study assessing whether the relative and absolute treatment effect of integrated AF care differed for individuals *depending on* their risk of all-cause mortality. Although using different methodology and observational registry data, the study by Romiti *et al.* identified a high and moderate clinical complexity cluster of patients, and showed that mortality was largely reduced among patients

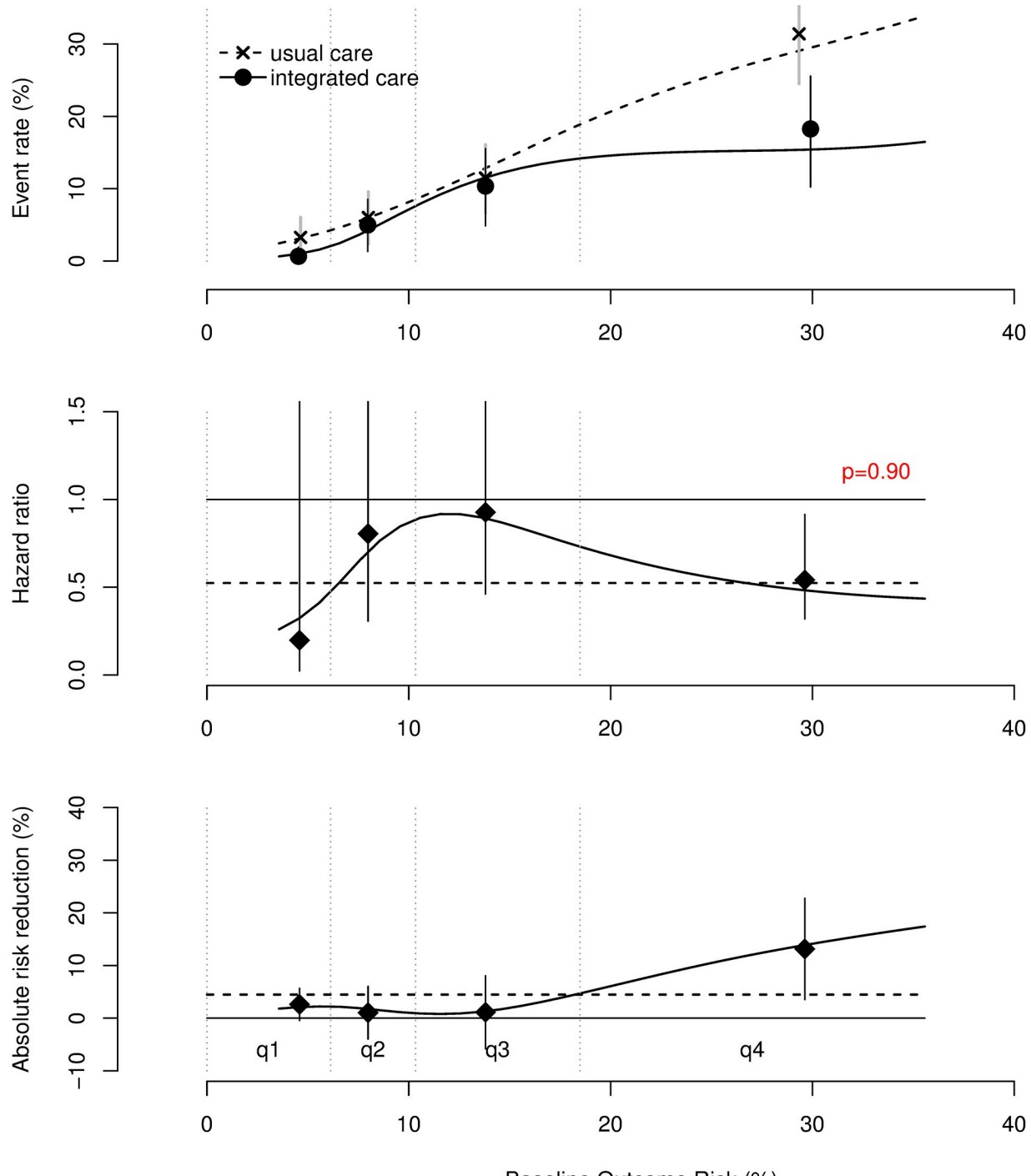

**Fig 1. Heterogeneous treatment effects analysis of integrated AF care in primary care setting based on ALL-IN study population.** *The event rate (top), the hazard ratios (middle), and the absolute risk reduction (bottom) are plotted as a function of the baseline outcome risk (i.e., predicted 2-year all-cause mortality risk). The intervention group (integrated AF care) is compared to usual care. The dashed line depicts the average effect (HR 0.55). q1, q2, q3 and q4 are four risk quarters. The vertical lines are 95% confidence intervals.*

with adherence to the integrated ABC-pathway (compared to non-adherence) in patients in the high clinical complexity cluster (HR 0.74 [95% CI 0.56–0.98]), *but not* in patients in the moderate clinical complexity cluster (HR 0.83 [95% CI 0.59–1.16]) [18]. A formal HTE analysis, with benefit expressed on an absolute scale, however, was missing. Other studies have reported conventional subgroup analyses evaluating integrated AF care. A study evaluating nurse-led AF care in a cardiology outpatient setting showed that the relative beneficial effect of the intervention was consistent over subgroups (e.g. patients with/without hypertension, or with/without heart failure) regarding the reduction of the composite outcome cardiovascular hospital admission or cardiovascular death, with the exception of females [3] for which the authors could not find an explanation. In the RACE-IV study, which was performed in a secondary and tertiary care setting, nurse-led AF care did not significantly reduce the risk of cardiovascular hospital admission or death compared to usual care provided by the cardiologist. Yet, an exploratory analysis showed that nurse-led AF care was effective in the subgroup of centers with experience in nurse-led care [7]. This could possibly be explained by higher guideline and study protocol adherence in experienced centers in the nurse-led care group. Although these conventional subgroup analyses may certainly be informative, their results should be interpreted with some caution. While patients are stratified according to one or two characteristics (e.g., sex and the presence or absence of concurrent heart failure), in reality, many more factors affect the (relative or absolute) effect of a treatment intervention. Our study therefore not only explores–for instance–sex as an explanation for differences in treatment benefit but combines this with many other important clinical characteristics. Additionally, unlike prior analyses, we also explicitly examine HTE on the clinically most meaningful scale, absolute risk difference.

## Interpretation of the findings

In general, an average overall effect (absolute and/or relative risk reduction) is reported in randomized trials. However, for optimal individualized decision-making, personalization of the treatment effect is more informative. Although in this study all AF patients seem to benefit similarly from integrated AF care on a relative scale, the absolute effect was the greatest in patients with a high predicted all-cause mortality risk, this is in patients in whom many clinical variables from the $CHA_2DS_2-VASc$ score are present. With increasing age, there is an accumulation of risk factors for all-cause mortality, due to ageing but also due to interacting comorbidities. These comorbidities along with polypharmacy, high biological vulnerability, dependency on significant others and a reduced capacity to resist stressors, together create frailty that so often is found in AF patients [19]. In fact, addressing all these components of frailty at once are the exact merits of integrated AF care. The results of our analysis of heterogeneity in treatment effect show that the higher the burden of comorbidities–and thus possibly the higher the frailty–the greater the effect of integrated AF care. This exemplifies the importance of addressing the 'C' component (detection and management of comorbidities) in the ABC approach.

## Clinical implications

Integrated AF care should not only address AF itself but also evaluate early signs (or worsening) of complications of AF, such as heart failure, as well as non-cardiovascular comorbidities [10]. As with any treatment, the associated benefit and burdens of integrated AF care should be weighed for individual patients to decide to whom to offer the intervention most intensely in everyday practice. Integrated AF care is not associated with harm and does not carry many burdens for patients, notably when it is organized close to their homes in primary care. Thus, in settings with limitless resources, it may be worthwhile to treat all patients with AF meeting

trial inclusion criteria with integrated care. However, in many settings characterized by resource constraints, prioritizing labor-intensive integrated AF care to those with the highest expected absolute effects seems a reasonable approach. Accordingly, we believe this is an important observation warranting further investigation: permanent AF in older, frail individuals certainly is not a stable 'cooled-down' disease. On the contrary, the all-cause mortality risk is high and an integrated cardiovascular care program, such as ALL-IN, has the largest absolute treatment effects precisely in this population. This knowledge may help to prioritize high-risk patients who might need stringent care, and to select low-risk patients, who might need less stringent care; a more lenient approach focusing perhaps more on self-management.

## Strengths and limitations

We used a state-of-the-art method to study heterogeneity of treatment effects, thus averting the disadvantages of conventional subgroup analyses. We were able to predict all-cause mortality based on often used and readily available clinical variables from the $CHA_2DS_2$-VASc score, with a good predictive performance upon external validation. Also, the results of the ALL-IN trial are highly generalizable, especially to the elderly, high-risk population that is typically present in primary care (median age ALL-IN trial was 77±11 years, versus 64±10 years in the secondary care population of the RACE 4 trial, for example). These strengths notwithstanding, some limitations should be considered. The results of this HTE analysis are dependent on the model used for risk prediction and thus on the selected predictors and outcome. It was not our primary aim to develop a model for formal use to predict mortality in future clinical practice, and we did not consider predictors beyond the clinical variables from the $CHA_2DS_2$-VASc score. Such additional variables beyond those included in the extensive model in the sensitivity analysis may predict mortality but were not (uniformly) available in both datasets. Nonetheless, the predictive performance of the model was good. This study focused on the outcome of all-cause mortality, yet other risks, such as the risk of ischemic stroke or the risk of hospital admission are also relevant for individualized decision-making in AF patients. Moreover, HTE analysis might be even more useful when applied in the analysis of individual data of multiple studies (IPD); by pooling results variation in the baseline outcome risk increases, and the statistical power is increased. In such circumstances, power might be sufficient to explore relative treatment effect modification of individual clinical variables—so called "effect modeling"—potentially uncovering even greater variation in treatment effects [12]. However, the ALL-IN trial was the first study to evaluate integrated AF care in primary care and so we did not explore treatment effect interactions beyond the that with risk, since this is likely to results in overfitting especially without strong prior information on relative effect modifiers [20]. Also, due to the cluster-randomization of the ALL-IN trial some imbalances were created between study arms that we did not correct for. However, these differences were minor, not univocally in favor of one study arm, showed no influence in the primary analysis of the ALL-IN trial [10], and, importantly, are taken into account when stratifying by the predicted mortality risk. We also note that we only considered the primary outcome all-cause mortality, and further research may focus on studying other outcomes. Finally, we have not performed a formal decision or cost-effectiveness analysis to suggest a threshold at which treatment may be attractive. This threshold might be expected to vary depending on resources available and the capacity of the clinics to offer this service.

## Conclusion

The relative degree of benefit from integrated care was shown to be similar in all AF patients managed in primary care in cooperative care with the cardiologist. Importantly, on an absolute

scale, the benefit was greatest in patients with a high predicted all-cause mortality risk, i.e., in frail older AF patients with multiple positive clinical variables from the $CHA_2DS_2$-VASc score items. These results help in efficiently organizing integrated AF care, expending limited resources more on high-risk patients and to a lesser extent on low-risk patients.

## Supporting information

**S1 Table. Patient characteristics of study population for model development and the ALL-IN study population.**
(DOCX)

**S2 Table. Development and internal validation of the prediction model.**
(DOCX)

**S1 Fig. Calibration plot of the internal and external validation of the prediction model.**
(TIF)

**S2 Fig. Distribution of predicted risk of all-cause mortality in total ALL-IN study population, intervention group and usual care group.**
(TIF)

**S3 Fig. Sensitivity analyses of heterogeneous treatment effects analysis of integrated AF care in primary care setting based on ALL-IN study population: Extended model.** The event rate (top), the hazard ratios (middle), and the absolute risk reduction (bottom) are plotted as a function of the baseline outcome risk (i.e., predicted 2-year all-cause mortality risk). The intervention group (integrated AF care) is compared to usual care. The dashed line depicts the average effect (HR 0.55). q1, q2, q3 and q4 are four risk quarters. The vertical lines are 95% confidence intervals.
(TIF)

**S4 Fig. Sensitivity analyses of heterogeneous treatment effects analysis of integrated AF care in primary care setting based on ALL-IN study population: CHA2DS2-VASc score as univariate predictor.** The event rate (top), the hazard ratios (middle), and the absolute risk reduction (bottom) are plotted as a function of the baseline outcome risk (i.e., predicted 2-year all-cause mortality risk). The intervention group (integrated AF care) is compared to usual care. The dashed line depicts the average effect (HR 0.55). q1, q2, q3 and q4 are four risk quarters. The vertical lines are 95% confidence intervals.
(TIF)

**S1 Appendix.**
(DOCX)

## Author Contributions

**Conceptualization:** Emmy M. Trinks-Roerdink, Geert-Jan Geersing, Carline J. van den Dries, Frans H. Rutten, Sander van Doorn.

**Formal analysis:** Emmy M. Trinks-Roerdink, Sander van Doorn.

**Methodology:** Maarten van Smeden, David van Klaveren, David M. Kent, Sander van Doorn.

**Supervision:** Geert-Jan Geersing, Martin E. W. Hemels, Isabelle C. van Gelder, Frans H. Rutten, Sander van Doorn.

**Writing – original draft:** Emmy M. Trinks-Roerdink, Geert-Jan Geersing, Carline J. van den Dries, Sander van Doorn.

**Writing – review & editing:** Emmy M. Trinks-Roerdink, Geert-Jan Geersing, Carline J. van den Dries, Martin E. W. Hemels, Michiel Rienstra, Isabelle C. van Gelder, Maarten van Smeden, David van Klaveren, David M. Kent, Frans H. Rutten, Sander van Doorn.

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
