## [Decision Letter · Decision Letter 0]

20 Jul 2023

PONE-D-23-16112Integrated care in patients with atrial fibrillation- a predictive heterogeneous treatment effect analysis of the ALL-IN trialPLOS ONE

Dear Dr. van Doorn,

Thank you for submitting your manuscript to PLOS ONE. After careful consideration, we feel that it has merit but does not fully meet PLOS ONE’s publication criteria as it currently stands. Therefore, we invite you to submit a revised version of the manuscript that addresses the points raised during the review process.

Please assess carefully all the reviewers comments.

We look forward to receiving your revised manuscript.

Kind regards,

Giulio Francesco Romiti

Academic Editor

PLOS ONE

“I have read the journal's policy and the authors of this manuscript have the following competing interests: G.J. Geersing, F.H. Rutten, and M.E.W. Hemels report unrestricted institutional grants for performing research in the field of atrial fibrillation from Boehringer-Ingelheim, Bayer Healthcare, BMS Pfizer and Daiichi Sankyo. I.C. van Gelder reports consultancy fees from Boston, BMS and Bayer to the institution, unrestricted research grants from the Netherlands Cardiovascular Research Initiative, unrestricted research grant from the European Union’s Horizon 2020 research and innovation programme under grant agreement: EHRA-PATHS (945260). M. Rienstra reports Consultancy fees from Bayer, Microport, InCarda Therapeutics to the institution, an unrestricted research grant from ZonMW and the Dutch Heart Foundation; DECISION project 848090001, unrestricted research grants from the Netherlands Cardiovascular Research Initiative: an initiative with support of the Dutch Heart Foundation; RACE V (CVON 2014–9), RED-CVD (CVON2017-11), an unrestricted research grant from Top Sector Life Sciences & Health to the Dutch Heart Foundation (PPP Allowance; CVON-AI (2018B017)), and an unrestricted research grant from the European Union’s Horizon 2020 research and innovation programme under grant agreement; EHRA-PATHS (945260). S. van Doorn reports an unrestricted institutional grant for performing research in the field of stroke diagnosis from Stoffels-Hornstra.”

Reviewers' comments:

Reviewer's Responses to Questions

**Comments to the Author**

1. Is the manuscript technically sound, and do the data support the conclusions?

Reviewer #1: Yes

Reviewer #2: Partly

2. Has the statistical analysis been performed appropriately and rigorously? 

Reviewer #1: Yes

Reviewer #2: Yes

3. Have the authors made all data underlying the findings in their manuscript fully available?

Reviewer #1: Yes

Reviewer #2: Yes

4. Is the manuscript presented in an intelligible fashion and written in standard English?

Reviewer #1: Yes

Reviewer #2: Yes

5. Review Comments to the Author

Reviewer #1: I have reviewed the manuscript entitled “Integrated care in patients with atrial fibrillation- a predictive heterogeneous treatment effect analysis of the ALL-IN trial" by Trinks-Roerdink et al.

This is a nice and interesting study demonstrating that AF patients at higher risk of mortality are actually those who benefit more from integrated and holistic care.

The manuscript is clear and well-written. From a methodological point of view, I do not see major issues or concerns.

However, I do not understand well the rationale of the developed a new prediction model for all-cause mortality. Indeed, it is not declared as one of the objectives of the study, and the explanation of this result in the Results section is very brief. In my opinion, it does not add too much to the study and probably deserves another paper where the authors should give more details.

Minor comments:

CHADS-VASc should be spelled out as CHA2DS2-VASc in the abstract, discussion, and conclusion.

Reviewer #2: The manuscript presented reports on a very interesting analysis regarding the impact of an integrated care strategy in reducing risk of all-cause death, according to predicted baseline risk in an HTE analysis.

The study surely poses an interesting question, focusing on a relevant clinical problem. Indeed, in this era in which limited resources are available (and personalised medicine is an aim), could be needed to identify more carefully patients to be treated according to certain standards, in order to maximize the effectiveness of clinical strategies.

The results showed are extremely interesting, since underlining the fact that patients with a higher predicted risk can gather the larger reduction in risk when treated with an integrated care approach is extremely important to facilitate the implementation of such clinical strategies.

Notwithstanding, I thing some issues are really important to be managed in the manuscript:

- I'm a bit concerned about the model used to evaluated the baseline predicted risk. The authors explained that a model was derivated from an external cohort, using the same factors part of the CHA2DS2-VASc score (but not the score itself, since they argue that was not originally validated for all-cause death) and then applied to the study cohort, to avoid biases related to the fact that the same factors could influence the choice of treatment. First of all, I don't see the same danger of bias related to the confounding-by-indication bias. If we consider that the original study was a cluster randomized trial, all the patients in each cluster received the same treatment approach, hence none of the clinical factors should have influenced it. Second, the cohort used to derive the model is almost ten years old and, very likely, including patients completely different from those included in this current study. Third, if we should use a different model, I would not use only those few clinical factors, but would enlarge the list including other important factors influencing the risk of death, such as CKD, COPD, Liver Disease; all these conditions have shown a direct independent influence on the risk of death in AF patients. Fourth, despite not being originally validated for all-cause death, there are solid proofs that CHA2DS2-VASc is able to predict all-cause death occurrence (doi: 10.1177/2047487318817662).

I would advise the authors, if they want to keep the current approach with the developed risk model (nonetheless I would consider whether to change the approach and develop the risk model in the current cohort, to then validate it externally), to show the baseline characteristics of the patients in the derivation cohort, at least to compare indirectly with the current cohort. They would maybe try to create a larger model, as a sensitivity analysis, including more clinical factors. Also, I would advise to perform the same analysis using the actual CHA2DS2-VASc score, considering the fact that is widely used in the clinical practice; this would allow to put their results more easily in the daily clinical practice. Such results would strenghten their main one.

- Another issue related to the predicted risk model is related to the calibration. The two plots (which actually I will incorporate in the main paper, instead avoiding Table 2 which is only little informative) looks like having completely different scales and then are impossible to be compared even visually, but even to make any assumptions. Indeed, by looking at the Supplementary Figure 2, it seems like the calibration is not so good in the lower risk group, but being the axis truncated at 0.6 is difficult to be completely sure about what we.

The authors should put both the curves on the same figure, not truncate any axes and use the same scale, including the figure in the main paper (also with a clear legend, that is currently missing). They should then comment and build up more on this point. Indeed, considering that the most of the patients have a low risk, is essential that the model perform at least acceptably in such patients.

- I think the authors could build up more about the impact of their results, regarding the issue of multimorbidity and frailty, which are very common in AF patients (DOI: 10.1016/j.lanepe.2022.100386, 10.1016/j.arr.2022.101652). How would their finding impact in the clinical management of such patients?

---

## [Author Response · Author response to Decision Letter 0]

11 Sep 2023

Please find a response to the reveiwers' comments in the uploaded files.

---

## [Decision Letter · Decision Letter 1]

26 Sep 2023

Integrated care in patients with atrial fibrillation- a predictive heterogeneous treatment effect analysis of the ALL-IN trial

PONE-D-23-16112R1

Dear Dr. van Doorn,

We’re pleased to inform you that your manuscript has been judged scientifically suitable for publication and will be formally accepted for publication once it meets all outstanding technical requirements.

Kind regards,

Giulio Francesco Romiti

Academic Editor

PLOS ONE

**Comments to the Author**

6. Review Comments to the Author

Reviewer #2: The authors addressed all the comments and the manuscript is now significantly improved. I found particularly interesting the reproducibility of the results in the two sensitivity analyses.

---

## [Editor Report · Acceptance letter]

12 Oct 2023

PONE-D-23-16112R1 

Integrated care in patients with atrial fibrillation- a predictive heterogeneous treatment effect analysis of the ALL-IN trial 

Dear Dr. van Doorn:

I'm pleased to inform you that your manuscript has been deemed suitable for publication in PLOS ONE. Congratulations! Your manuscript is now with our production department. 

Kind regards, 

on behalf of

Dr. Giulio Francesco Romiti 

Academic Editor

PLOS ONE